# International medical graduates' experiences of clinical competency assessment in postgraduate and licensing examinations: A scoping review

Helen Hynes[1]*, Nora McCarthy[1], Anél Wiese[1], Catherine Sweeney[1], Nitin Gambhir[2], Tony Foley[3‡], Deirdre Bennett[1‡]

1 Medical Education Unit, School of Medicine, University College Cork, Cork, Ireland, 2 Lead Dean Director, Public Services Delivery Scotland, Glasgow, Scotland, 3 Department of General Practice, School of Medicine, University College Cork, Cork, Ireland

☉ These authors contributed equally to this work.
‡ TF and DB authors also contributed equally to this work.
* h.hynes@ucc.ie

## Abstract

### Background

International Medical Graduates (IMGs) represent a substantial proportion of the medical workforce globally and are known to face difficulties in their working lives, including differential attainment in assessment. This scoping review explores existing literature relating to IMGs' experiences of clinical competency assessments and identifies gaps in current knowledge.

### Methods

Following the Arksey and O'Malley framework for scoping reviews we examined peer-reviewed literature published between 2009 and 2025, without language restrictions. Three independent reviewers contributed to each stage of the process. The British Education Index, the Education Resources Information Center (ERIC), PubMed, Psych Info, Scopus, and Soc Index were searched. Forwards and backwards citation searching and a grey literature search were also performed.

### Results

We identified 44 publications which met the inclusion criteria. These described the experiences of over 7200 IMGs, who had received their primary medical qualifications in 54 named countries, spanning all 6 world regions as defined by the World Health Organisation. Most sources were from the United Kingdom, Australia, and Canada. The review identified three headline categories which summarise the factors that impacted on IMGs' assessment experiences: internal and personal factors; external and social factors; and institutional and systemic factors. Notable evidence gaps included the impact of age, gender, country of primary medical qualification, years

**Data availability statement:** All relevant data are within the manuscript and its Supporting Information files.

**Funding:** The author(s) received no specific funding for this work.

**Competing interests:** The authors have declared that no competing interests exist.

since last in practice, and lower levels of language proficiency on IMGs' experiences of assessment. Our review reveals the need for further research from a broader range of medical specialties and from a wider geographic spread to improve our understanding of this important topic.

## Conclusion

This scoping review maps the experiences of IMGs in relation to clinical competency assessment and offers useful insights highlighting both positive and negative influences, which could inform future assessment methodologies and support interventions for IMGs. By creating fairer and more supportive assessment pathways, healthcare systems can harness the full potential of IMGs, ensuring their valuable skills and diverse experiences are retained for the benefit of all patients.

## Background

International medical graduates (IMGs) are defined as medical doctors who are practicing in a country that is not the same as that in which they received their primary medical qualification [1]. Many healthcare systems rely heavily on the work of IMGs, who make up around 25% of doctors in the United States of America (USA), over 30% in Australia, 39% in the United Kingdom (UK) and 40% in Ireland [2–5] and this is projected to rise further in the near future [6,7].

IMGs are sometimes required to pass licensing examinations before they can register to work. In some countries, like the USA, this is the same examination required of locally trained doctors [8]. In others, such as Ireland, the UK, Australia and New Zealand, IMGs must take examinations specific to doctors who have qualified abroad, although IMGs from certain countries may be exempted from this requirement [9–12]. Additionally, specialty training completed abroad may not be recognised, requiring IMGs to retrain or pass postgraduate certification examinations to be acknowledged as specialists. This review considers both licensing and postgraduate specialty examinations.

Typically, licensing and postgraduate examinations contain tests of both knowledge and clinical competency. While knowledge tests primarily assess a doctor's theoretical understanding, clinical competency assessments are designed to evaluate the application of knowledge and related clinical skills in a practical context and may include Objective Structured Clinical Examinations (OSCEs), and direct observation of skills in workplace settings [13,14].

There are significant differences between the experience of undergoing clinical competency assessment and that of sitting for knowledge-based written examinations [15]. Clinical examinations involve complex dynamics between candidates, examiners, real patients, and simulated patients, making them a more authentic measure of competency than is possible in purely knowledge-based assessments [16,17].

A widely documented issue in medical education is the existence of differential attainment among IMGs undertaking medical licensing and postgraduate

examinations [18–27]. This is the phenomenon whereby individuals experience different outcomes (in relation to assessments and opportunities) based on to their age, race, gender, sexual orientation, ethnicity, disability, socio-economic deprivation or migrant status rather than on their ability, effort or motivation [28]. Differential attainment is a widespread phenomenon in education [29] including in the field of medical education where it has been described in both knowledge and clinical competency tests [26,30] and persists despite correcting for possible confounders [31]. It is documented in many different specialties including Intensive Care Medicine [22], Paediatrics [32], Obstetrics and Gynaecology [33], Radiology [34], Surgery [19,23], Oncology [35] and General Practice [21,25–27,36,37].

Given the significant reliance on the work of IMGs to support many different healthcare systems, and the well documented evidence of differential attainment in assessments, we wanted to explore what is currently known about IMGs' experiences of clinical assessment. The focus of this review is on clinical competency testing, rather than knowledge testing, because, as outlined above, clinical assessments more closely reflect the realities of everyday practice of medicine, making them a highly relevant measure of an IMG's ability to adapt and integrate into a new health-care system [38,39].

Before starting this scoping review, a preliminary search was conducted in PubMed, JBI Evidence Synthesis and the Cochrane Database of Systematic Reviews to identify any existing reviews on the topic of IMGs' experiences of clinical assessment. A rapid review, commissioned by the General Medical Council in 2015 to investigate differential attainment in medical training pathways found that most studies concentrated on examination outcomes, such as pass/fail rates and progression or non-progression, while overlooking the experiences, perspectives, and attitudes of IMGs regarding assessments [40]. To the best of the authors' knowledge no reviews have been carried out on IMGs' experiences of clinical competency assessment. Therefore, the impetus for this scoping review arose from the need to systematically map the evidence base relating to IMGs' lived experiences during clinical competency tests, such as OSCEs, workplace-based assessments, direct observation of procedures, and mini-clinical examinations.

This scoping review aims to:

(1) map key concepts and evidence from academic and grey literature regarding the experiences of IMGs with clinical competency assessments; and

(2) identify gaps in current knowledge of this topic through systematic searching, selection, analysis, and synthesis and thus to guide future research.

## Methodology

This review was conducted in accordance with a pre-established protocol that was registered on the Open Science Framework (available at https://osf.io/8gdm7) and published in a peer reviewed journal [41].

A scoping review was chosen for its effectiveness in summarizing existing knowledge and identifying gaps. The process followed Arksey & O'Malley's framework [42] and the findings are presented using the PRISMA ScR format [43,44]. See S1 Appendix.

### Identifying the research question

The primary research questions were:

• What literature has been published relating to the experiences of international medical graduates undertaking clinical postgraduate and licensing medical examinations; and

• What experiences do international medical graduates describe in relation to clinical postgraduate and licensing medical examinations?

The secondary research question was:

- What are the gaps in the literature relating to our knowledge and understanding of international medical graduates' experiences of clinical postgraduate and licensing medical examinations?

### Identifying relevant sources for inclusion

To develop an effective and comprehensive search strategy, an initial limited search of PubMed and Scopus was conducted to identify relevant search terms and medical subject headings (MeSH). Then a comprehensive search strategy was devised with the assistance of a research librarian and adapted for each database. To ensure broad coverage across medical, health, and educational disciplines the following databases were searched: British Education Index, ERIC, Psych Info, PubMed, Scopus, and Soc Index. A full search strategy for one of the databases can be viewed in Appendix 2a in S2 Appendix. The original database searches were performed on May 10th, 2024, and were re-run on November 1st, 2025, to capture the most recent publications in the field. A grey literature search was carried out using Google on Oct 24th, 2024, and again on November 2nd, 2025. The grey literature search strategy can be viewed in Appendix 2b. Reports from relevant international stakeholders, such as postgraduate medical training bodies and medical licensing organizations were searched. A forward and backwards citation search and a hand search of subject experts were also conducted.

### Study selection

All identified sources were collated and uploaded into Covidence systematic review software [45]. Duplicates were removed. Inclusion and exclusion criteria were developed using the headings Population, Concept and Context [43] as seen in Table 1 below.

Publications from 2009 to 2025, which were available in full text, were considered for inclusion. This date range was chosen because postgraduate and licensing examinations undergo regular reviews and updates, making experiences from versions of these assessments from before this period less relevant to the current challenges faced by IMGs. As the aim for the review was to capture what is known in an international context, publications in any language were considered.

**Table 1. Inclusion and Exclusion criteria.**

|  | Inclusion | Exclusion |
|---|---|---|
| Population | Data source relating to International Medical Graduates (medical doctors) | Sources relating to locally trained medical graduates. Sources which do not differentiate where the graduates trained. Sources relating to other health care professionals (not doctors). |
| Concept | Sources relating to the experiences of the participants | Sources which do not refer to the experiences of participants |
| Context | Sources relating to postgraduate medical examinations. Sources relating to licensing or credentialing medical examinations. Sources relating to clinical competence assessment (including but not confined to OSCEs, Workplace based assessments (WBA), Direct Observation of Procedures, Mini-Clinical Examinations) | Sources relating to undergraduate medical examinations. Sources relating only to knowledge-based assessment such as Multiple-Choice Questions (MCQs) |
| Types of sources | Qualitative, quantitative, or mixed methods studies; 'grey' literature such as reports, reviews, theses, letters, book chapters, opinion pieces, and organisational documents. Published between 2009–2025. | Published prior to 2009 |

At all stages of the process, whenever the reviewers disagreed on a publication, it was re-examined through discussion and consensus-building until agreement was reached.

Three reviewers jointly screened the first 20 sources to pilot the inclusion and exclusion criteria and to ensure their clarity and usability. Subsequently, the same three reviewers independently screened the titles and abstracts of all citations using the inclusion and exclusion criteria with each item being screened by two of the three reviewers. Thereafter, full texts were retrieved for sources that were identified as potentially relevant, and these were assessed in detail against the inclusion criteria by two of the three independent reviewers. Sources that did not meet the inclusion criteria for full-text review were removed. The reasons for exclusion were recorded and are reported in the results section below. The search results and study inclusion process are presented in Fig 1 below, utilising the PRISMA-ScR (Preferred Reporting Items for Systematic Reviews and Meta-Analyses extension for Scoping Reviews) flow diagram [44].

### Data extraction

Once papers had been selected for inclusion, data was extracted using a data extraction tool developed by the reviewers (see S3 Appendix). To ensure effectiveness and reliability, the tool was piloted by three reviewers working together for the first four sources and then in pairs for the next four. This piloting phase involved comparing extracted data to establish initial inter-rater reliability and refine the tool before independent use. The remaining data extraction was carried out by the principal investigator (HH) and was reviewed for accuracy and consistency by the other members of the research team. The data extracted included specific details about the participants (IMGs), the concept (the experiences described by the IMGs), the context (in relation to clinical competency assessment), study methods and key findings relevant to the review questions.

### Collating, summarizing and reporting of the results

Tables were used for description of study characteristics. NVIVO 15 [46] was used to extract information from the sources. Ideas were then grouped and organised into categories and subcategories, based on similarity of content, following consensus among members of the research team.

## Findings

### Research question 1: What literature has been published relating to the experiences of international medical graduates undertaking clinical postgraduate and licensing medical examinations?

The database search identified 1355 sources, with 6 further sources from grey literature, 10 from forward and backwards citation searches, and 1 from hand searching, totalling 1372 sources which were imported into the Covidence software platform [45] for screening. Covidence identified 457 duplicates which were removed, leaving 915 sources for screening. Of these 808 were excluded as irrelevant leaving 107 sources for the full text evaluation. A further 63 were excluded based on the search criteria for the reasons listed in the PRISMA diagram (Fig 1), leaving 44 sources for the final review.

### Description of the sources included in the scoping review

Publication output varied over the 15-year period with peaks occurring in 2010, 2012 and the highest number of sources (18%, n = 8) being from 2020.

The included sources were published across 12 countries. The largest proportion originated from the United Kingdon (32%, n = 14), followed by Australia (30%, n = 13), and Canada (11%, n = 5). Sources from the USA accounted for 7% (n = 3), while combined sources from Australia and New Zealand represented 4% (n = 2). A further 4% (n = 2) were from the Netherlands and single studies (2% each) were identified from Chile, Finland, Germany, and Sweden. One source (2%) reported data derived from multiple countries.

Of the 44 sources included, 57% (n = 25) were qualitative, 11% (n = 5) were quantitative, 30% (n = 13) were mixed methods and 1 (2%) was a short report which did not describe the methods used. The sources were predominantly peer

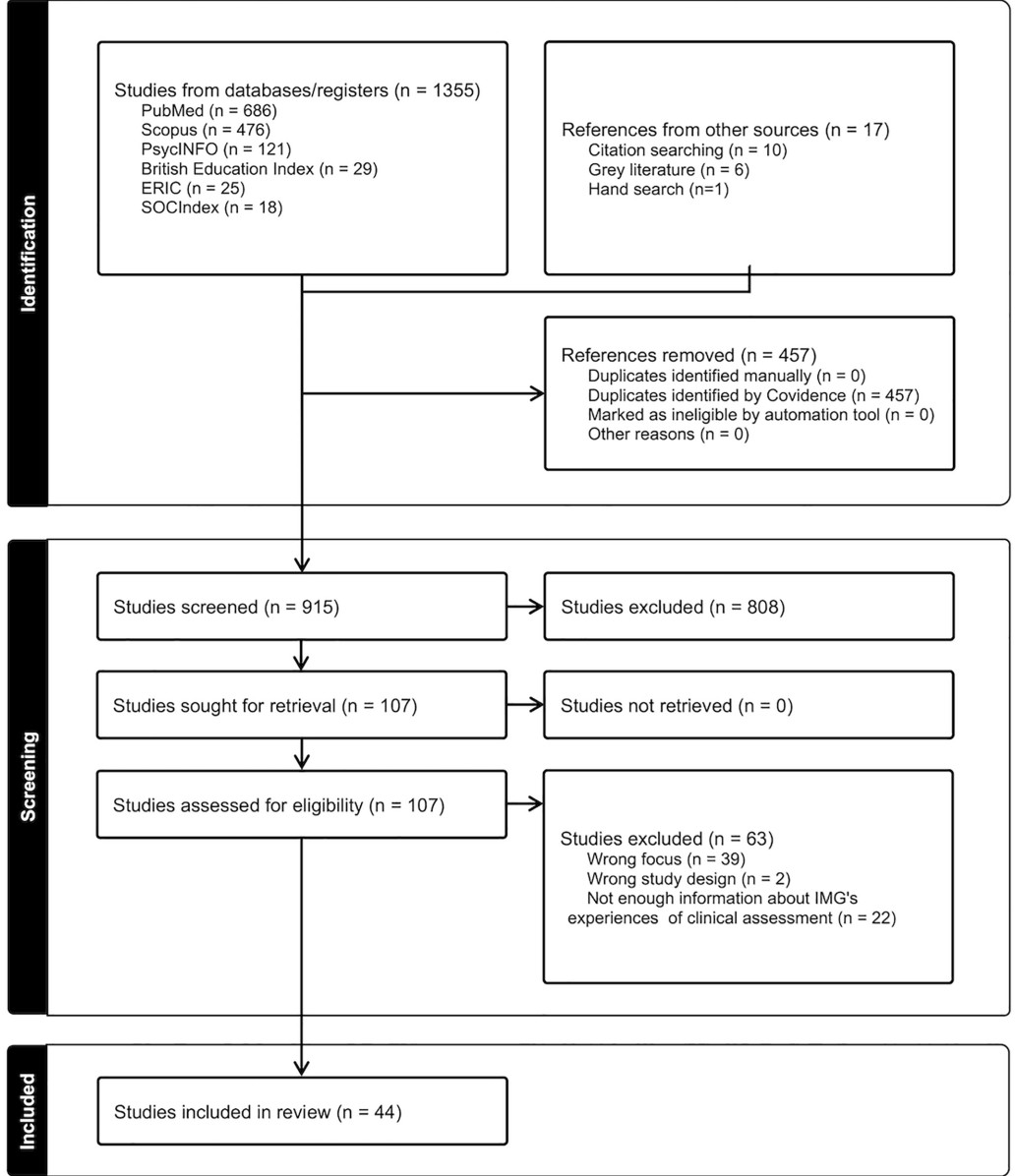

**Fig 1. PRISMA-ScR Flow Diagram.**

reviewed empirical papers (77%, n = 34). There were also 3 reports (7%), 3 theses (7%), 1 letter (2%), 1 conference abstract (2%), and 1 blog post from the GMC website (2%).

All 44 related to IMGs' experiences of clinical assessment. Licensing or accreditation assessments accounted for 52% (n = 23) of sources. This included licensing examinations in Australia (25%, n = 11), Canada (9%, n = 4), USA (4%, n = 2), UK (4%, n = 2), Netherlands, Finland, Sweden and Germany (2%, n = 1 each).

Post-graduate specialty assessment accounted for 39% of the sources (n = 17) and 9% (n = 4) described IMGs undergoing both licensing and postgraduate assessment.

General Practice (GP) was the most frequently studied specialty (18%, n = 8), followed by psychiatry (9%, n = 4) oncology (2%, n = 1) and paediatrics (2%, n = 1). The remaining publications either did not specify a specialty or contained a mixture of specialties including GP, Psychiatry, Anaesthesiology, Oncology, Paediatrics, Obstetrics & Gynaecology, Internal Medicine, Neurology, Emergency Medicine, General Surgery, Cardiothoracic Surgery, Urology, Orthopaedics, Haematology, Laboratory Medicine and Public Health.

Forty-three of the studies were in English. One was published in Spanish (87) and this was translated by one of the authors (HH). See Table 2 below. Further details on all included sources are outlined in S4 Appendix.

Table 2 below summarises each of the sources included in this scoping review.

## The study participants

Approximately 7200 IMGs were included in the 44 sources. It was not possible to calculate an exact number because 2 publications did not give the number of participants [20,47] and the largest study, a 2024 report from the Australian Medical Council [3] listed the number of participants as *"more than 4000"*. However, from the data we have we can say that there were at least 4564 IMGs in the qualitative sources, 252 in the quantitative sources and 2415 in the mixed sources.

Amalgamating the participants from all 44 publications, IMGs had received their primary medical qualification in 55 named countries spanning all 6 world regions as defined by the World Health Organisation [48] as seen in Table 3 below.

## Research question 2: What experiences do international medical graduates describe in relation to clinical postgraduate and licensing medical examinations?

The second objective of the review was to map how IMGs describe their experiences with clinical postgraduate and licensing assessments. These findings were grouped based on similarity into the following categories: internal and personal factors, external and social factors, and institutional factors related to assessment design and implementation.

## Category 1: Internal and Personal Factors

This category captures internal and personal factors relating to the IMGs themselves, which were documented in the sources, and which shaped their experience of clinical assessment. These include the burden of personal and financial practicalities, communication challenges, and the impact on wellbeing.

## Personal and financial practicalities

*"I'm poor. Part of my income comes from the salary of my wife, and the other part comes from social benefits. […] So it's … it's making me angry. It's making me nervous. It's making me frustrated because it's not how things are supposed to be. My wife shouldn't be going to work to clean people's houses while I'm sitting at home, studying for exams. That's not how it's supposed to be for medical specialists"* IMG – [49].

Participants described how family responsibilities impacted on their ability to devote themselves to their studies; this was particularly an issue for female IMGs who described how disproportionate childcare responsibilities impeded study time [50–55]. The cost of preparation courses, examination fees and travel were all cited as barriers to assessment for IMGs particularly when they had to balance these costs with their responsibilities to support their families [3,49,51,52,55–61]. In one publication, female IMGs who were able to study for examinations could do so only because their partners earned enough money to be able to support the family [52]. The perception of the examination process as a "diamond mine" generating significant profit for licensing bodies from IMGs, was also expressed [52].

**Table 2. Overview of publications included in the scoping review.**

| First Author and year | Publication Type | Type of Source | Country | # of IMG participants | Level of assessment | Type of assessment |
|---|---|---|---|---|---|---|
| Vamos 2009 | Quantitative | Peer reviewed journal article | Australia and New Zealand | 116 | Post-grad only | Fellowship of RANZCP (Psychiatrists) |
| Bourgeault 2010 | Qualitative | Report | Canada | 67 | Licensing only | MCC NAC OSCE |
| Higgins 2010 | Quantitative | Peer reviewed journal article | Australia | 73 | Post-grad only | ANZCA Final exam (Anaesthesiology) |
| Huijskens 2010 | Qualitative | Peer reviewed journal article | Netherlands | 32 | Licensing only | CIBA Programme at Maastricht University |
| Remedios 2010 | Other: Unclear what was used as not described | Peer reviewed journal article | UK | Not specified | Post-grad only | Membership of RCGP CSA |
| Webster 2010 | Mixed | Peer reviewed journal article | Australia | 91 | Post-grad only | Fellowship of RACS (Surgery) |
| Jamieson 2011 | Qualitative | Peer reviewed journal article | UK | 15 | Post-grad only | Membership of RCGP CSA |
| Tipton 2011 | Qualitative | Book | United States | 19 | Licensing only | USMLE Clinical Skills |
| Huthwaite 2012 | Qualitative | Peer reviewed journal article | Other: Australia and New Zealand | 32 | Post-grad only | Fellowship of RANZCP (Psychiatrists) |
| McGrath 2012 | Qualitative | Peer reviewed journal article | Australia | 30 | Licensing only | AMC clinical |
| Nair 2012 | Mixed | Peer reviewed journal article | Australia | 22 | Licensing only | WBA run by the University of Newcastle (Australia) as a trial alternative to the AMC Clinical Assessment |
| Rao 2012 | Quantitative | Peer reviewed journal article | United States | 20 | Post-grad only | Clinical Skills Verification Process (Psychiatry) |
| Slowther 2012 | Qualitative | Peer reviewed journal article | UK | 116 | Licensing only | PLAB |
| Low 2013 | Quantitative | Peer reviewed journal article | UK | 10 | Post-grad only | CSA – mock test run by Dorset Deanery |
| Peters 2013 | Qualitative | Thesis | Canada | 15 | Licensing only | CEHPEA QE1 and CE1 – Centre for the evaluation of health professionals educated abroad – comprehensive clinical examination – CE1 |
| Kuusio 2014 | Mixed | Peer reviewed journal article | Finland | 12 qual and 553 quant | Licensing only | Finish licensing exams - |
| Terry 2014 | Mixed | Peer reviewed journal article | Australia | 22 qual and 105 quant | Licensing only | AMC alternate WBA pathway |
| Kunakov 2015 | Quantitative | Peer reviewed journal article | Chile | 33 | Licensing only | Unique National Exam of Medical Knowledge (EUNACOM) – theoretical and practical exam including OSCE |
| Legido-Quigley 2015 | Qualitative | Peer reviewed journal article | UK | 23 | Post-grad only | GP exams and others |
| Nair 2015 | Qualitative | Peer reviewed journal article | Australia | 26 | Licensing only | AMC WBA |
| Ragg 2015 | Qualitative | Peer reviewed journal article | UK | 10 | Post-grad only | MRCGP CSA |
| Ayonrinde 2016 | Qualitative | Other | UK | 1 | Licensing only | PLAB |

*(Continued)*

| First Author and year | Publication Type | Type of Source | Country | # of IMG partici-pants | Level of assess-ment | Type of assessment |
|---|---|---|---|---|---|---|
| Woolf 2016 | Qualitative | Report | UK | 21 | Post-grad only | UK Royal College exams – various |
| Woolf 2016 | Qualitative | Peer reviewed journal article | UK | 36 | Post-grad only | UK Royal College exams – various |
| Moneypenny 2018 | Qualitative | Thesis | Canada | 12 | Licensing only | NAC OSCE |
| Khan 2019 | Mixed | Peer reviewed journal article | UK | 401 | Post-grad only | Induction and Refresher Scheme Simulated Surgery Examination |
| Cunningham 2020 | Qualitative | Letter | UK | 7 | Post-grad only | RCGP CSA |
| Harris 2020 | Qualitative | Peer reviewed journal article | Australia | 13 | Licens-ing and post-grad | AMC Clinical and various Australian post grad |
| Heist 2020 | Qualitative | Peer reviewed journal article | United States | 35 | Licensing only | USMLE |
| Leung 2020 | Qualitative | Peer reviewed journal article | Australia, Canada, Netherlands, China, Switzerland | 3 | Licensing only | Multiple |
| Loss 2020 | Qualitative | Peer reviewed journal article | Germany | 20 | Licensing only | German Licensing exams |
| Sood 2020 | Mixed | Thesis | Canada | 4 qual and 31 quant | Licensing only | Canadian licensing and NAC |
| Sturesson 2020 | Mixed | Peer reviewed journal article | Sweden | 14 | Licensing only | Swedish Licensing exam |
| Terry 2020 | Qualitative | Peer reviewed journal article | Australia | 4 | Licensing only | AMC alternate WBA pathway |
| Parvathy 2021 | Mixed | Peer reviewed journal article | Australia | 153 | Licensing only | AMC alternate WBA pathway |
| Mbeledogu 2022 | Mixed | Peer reviewed journal article | UK | 16 | Post-grad only | Virtual MRCPCH Clinical Examinations |
| Postmes 2023 | Qualitative | Peer reviewed journal article | Netherlands | 17 | Licensing only | Dutch licensing exam (AP) – Clinical Skills exam – OSCE |
| Rashid 2023 | Qualitative | Peer reviewed journal article | Canada | 10 | Licensing only | Canadian Licensing exams |
| Australian Med-ical Council 2024 | Qualitative | Report | Australia | 4000 approx. | Licens-ing and post-grad | AMC Clinical Exam and others |
| Hodge 2024 | Qualitative | Peer reviewed journal article | Australia | N/A | Licensing only | AMC Clinical Examination |
| Iyizoba-Ebozue 2024 | Mixed | Peer reviewed journal article | UK | 13 | Post-grad only | FRCR (Clinical Oncology) |
| Siriwardena 2024 | Mixed | Peer reviewed journal article | UK | 683 | Post-grad only | MRCGP |
| Healey 2025, | Mixed | Peer reviewed journal article | Australia | 286 | Licens-ing and Post-grad | AMC Clinical Exam and others |
| Healey 2025, a | Mixed | Peer reviewed journal article | Australia | 286 | Licens-ing and Post-grad | AMC Clinical Exam and others |

**Table 3. Countries of primary medication qualifications of participants in included sources.**

| Africa | Americas | Eastern Mediterranean | Europe | SE Asia | Western Pacific |
|---|---|---|---|---|---|
| Ghana | Antigua | Afghanistan | Belarus | Bangladesh | Australia |
| Mauritius | Canada | Egypt | Bulgaria | India | China |
| Nigeria | Columbia | Iran | Estonia | Indonesia | Japan |
| South Africa | Guatemala | Iraq | France | Myanmar | Malaysia |
| Zimbabwe | Mexico | Jordan | Germany | Sri Lanka | New Zealand |
| | Nicaragua | Lebanon | Greece | | Philippians |
| | Peru | Libya | Hungary | | South Korea |
| | Trinidad | Pakistan | Ireland | | |
| | USA | Sudan | Malta | | |
| | | Syria | Poland | | |
| | | UAE | Portugal | | |
| | | | Romania | | |
| | | | Russia | | |
| | | | Slovenia | | |
| | | | Spain | | |
| | | | UK | | |
| | | | Ukraine | | |
| | | | Uzbekistan | | |

## Communication challenges

*"There isn't time to let patients talk in the exam, so I will just memorize what to say and do all the talking"* [62].

Most publications reported that communication challenges experienced by IMGs impacted on their examination performance [35,47,49,51–53,57, 62–79]. IMGs demonstrated varying levels of familiarity with the language of their host countries: from being native speakers, to having received an immersive education in the language of their host country, to having received their primary medical qualification in the language of the host country, to having learned it only in secondary education or even as adults [72]. Even when they were very familiar with the language of their host country, IMGs still struggled with medical abbreviations [57], professional medical communication [51,57,63], colloquialisms [62,66,78] and how to use humour and informality in communication [66,71]. A specific challenge highlighted in one study was the overly difficult academic language used in examination items, leading to 'linguistic confusion' despite IMGs reporting no practical language issues in their daily work [53]. One study recommended providing IMGs with courses in public speaking to help them reduce their accents [74].

Problems interpreting body language and other non-verbal cues such as eye contact, tone of voice, facial expressions and gestures were also identified [73,76].

In one study IMGs believed that they were perceived as having poor clinical skills when the issue was, in their opinion, a language deficit [66]. The need to do language courses, and the difficulty accessing such language courses was mentioned as an impediment [67,68]. Conversely some IMGs chose to forego language training even when it was free and readily available, choosing instead to spend their time on medical related study, which they hoped would increase their chances of passing the licensing examination [72].

**Impact on wellbeing**

*"I am going through hell because my education, training, and work as a physician are not recognized here [...] I came from a Third-World country where I never experienced what I have experienced here. I am left empty, and I am so depressed that I cry every night thinking about what I have become and what I wanted to be in Canada"* – IMG - [60].

The process of trying to pass licensing and postgraduate examinations was documented as causing the IMGs to experience many negative emotions. The prospect of potential failure led to anxiety, insecurity, self-doubt, the worry of compounding financial difficulties and even having to abandon their dreams [3,35,52,53,56,57,59,62,64,78,79]. This was compounded by their awareness that they were statistically more likely to fail than locally trained graduates [3,53,66,73,78,79]. IMGs who had already failed these examinations described how their confidence was knocked [35,66,79]. One IMG said that potential study partners were no longer interested in practicing with him when they learned that he had previously failed the exam [62]. Some IMGs were worried that they might lose their visa status or be deported if they performed poorly in examinations [62,74].

Many sources described mental health issues experienced by IMGs in relation to assessment including depression, hopelessness and sleep disturbance [3,49,53,55,57,59,60,62,63,73,76–81]. Several sources also documented IMGs' feelings of humiliation, resentfulness, anger and frustration in response to the assessment process [3,49,53,57,60,72].

**Category 2: External and social factors**

This category documents the external and social influences, identified in the included sources, that impacted on IMG examination performance. Employment challenges, relationships, and models of practice are identified as factors in this category.

**Employment challenges**

*"I feel like a fish that is outside the water. And if I start working it will be like you threw a dying fish back in the water, literally giving me breath again. (…) Being a physician in Syria is not a job. It's an identity. So, you can imagine that the Syrian physician here is not looking for a job, he is looking for the identity"* – IMG – [49]

Employment played both positive and negative roles in the IMG journey. Some sources described how it was not possible for IMGs to get any kind of work, observership or experience in their new healthcare systems without which it was very difficult for them to acclimatize to the accepted models of practice, often leading to poor examination outcomes [49,60,67,72,80].

Nine sources reported IMG dissatisfaction that their prior training was not recognized [3,49,60,64,68,72,77,82,83] meaning that they needed to retrain. In many cases, doctors missed out on work or training opportunities because of their IMG status, with preference being given to locally trained graduates who were more likely to be in training positions [35,54,60,72,78,79].

A number of sources reported that the process of completing the required examinations could take several years, resulting in a prolonged period without any clinical practice. During this period, IMGs often needed to take non-medical jobs, leading to feelings of losing professional relevance, forgetting previously learned knowledge, and dwindling motivation [3,49,53,55,57,62].

Those who secured medical positions described how their heavy workload impeded their ability to study [50,54,65,70,75,77,79]. In some cases, IMGs who were preparing for postgraduate examinations found that the only medical positions they were able to obtain were service posts which were unaccredited for training [3].

### Relationships

*"Oh, you must do everything possible to get the locals [locally trained graduates] in. Bother them, haunt them, disturb them, anything [laughs]. Just get them in"* – IMG – [73].

Support from a trainer and other work colleagues was deemed very valuable when preparing for examinations [35,53,54,60,61,65,66,69–71,73,76,78,79]. Conversely, when trainers were not supportive IMGs felt demeaned, for example when consultants referred to WBA as a "soft" or "easy" option compared to the AMC clinical examination process [71]. One study reported that IMGs worried that they might be stigmatised by their trainers if they needed time off to attend courses [79].

Peer study groups were listed as very helpful for exam preparation [20,58,62,66,69,73]. Preparing with other IMGs was a commonly mentioned source of help [35,52,58,61,63–65,73,78–80,84]. Conversely some sources highlighted the benefits of studying with locally trained graduates [53,73,79] and one IMG described this as *"central for success"* [53].

Community integration, for example due to having children, was also considered beneficial as it helped to foster the development of cultural competency and communication skills [53,73,78].

IMGs who described feeling socially and/ or geographically isolated believed that this impeded their assessment progress [20,49,50,52–54,62,71,73,78,85]. Those who had not developed social networks found the assessment process particularly difficult [52]. Some sources described how the use of technology supported their training both with supervisors and with other IMGs [62,73].

### Models of practice

*"I think it's not fair [for] people who come from a different system to expect them to just merge into the system. in terms of you know understanding and the difference in language, the culture and so they need more time and more support"* – IMG – [35].

Models of medical practice in home countries differed from those in host countries and this contributed to the challenges experienced by IMGs in clinical assessment [49,55,59,60,62,66,72,73,79]. Some IMGs explained that they had been trained to practice in a doctor-centred manner and that this was not seen as acceptable in their host countries [63,64,66,76,83]. They felt *"uncomfortable"* with the patient-centred models of care that were expected of them and made the point that *"patients' expectations of a 'good doctor' are very different in different parts of the world"* [66]. One study found that when IMGs worked together in small groups, they often reinforced doctor-centred and biomedical models of the consultation. As a result, the authors recommended forming mixed study groups that included both IMGs and locally trained graduates [66].

One study mentioned that medical students in North America have a role in patient care whereas in their home country medical students are largely observers and this impacted upon their practice readiness and examination performance [59].

Several sources referred to the value of bridging programmes to help IMGs acclimatise to unfamiliar models of practice which would make it easier for them to pass their examinations [3,57,58,60,66,79]. One study referred to IMGs as feeling obliged to enrol on expensive courses to help them to learn about the hidden curriculum of how to practice in their host country [72]. Another study found that IMGs had a low attendance rate at residential study courses because they perceived that the main purpose of the course was for social and cultural purposes [47].

Cultural differences in practice were seen as a barrier for IMG success in assessments: [49,52,55–57,62,71–74,79]. IMGs highlighted cultural differences such as the way they dressed ([35], wearing a hijab [80], the appropriate way to

address colleagues [71] and when to shake hands with a patient [49] which they felt impacted on their examination performance. One study identified that IMGs lacked understanding of legal and professional frameworks, including the concepts of autonomy and confidentiality, having trained in cultures in which sharing information and involving family in decisions about patients were considered the norm [76].

**Category 3: Institutional factors that influence IMGs' experiences of assessment**

This category describes the institutional issues related to design and administration of the clinical competency assessments and the IMGs' perceptions thereof. Issues related to teaching, learning and assessment; challenges cause by bureaucracy; comments on feedback; and observations related to fairness were all documented in the sources and included in this category.

**Teaching, learning and assessment**

*"We don't learn or do this* [OSCE type assessment] *in my country, so how can I be expected to know it?"* – IMG – [62].

A common experience reported across many sources was the disparity between methods of teaching, learning, and assessment in IMGs' home country compared to those in their host country [47,54,59,60,62,72,74,75,79,86]. Some participants were unfamiliar with the required clinical assessments such as OSCEs [72], simulated patient encounters [62], and the use of portfolios and workplace-based assessment [75]. Testing of certain cultural domains such as sexual health, mental health, domestic violence, or social advocacy issues was alien to some IMGs [62,72].

IMGs found it difficult to source information regarding assessments which made it difficult for them to prepare adequately [3,50–52,54,55,58,60,65,67,72,77,80,82,84,87]. An IMG described the AMC Clinical examination process as *"horrendous"* and *"faulty"* because *"it expects IMGs to function like Australian trained doctors yet fails to provide adequate training and resources (including humans to talk to) for IMGs to get any idea of what it is expected of them"* [3]. This concept was echoed by sources in Canada where IMGs were unable to practice until they passed the licensing examination but without any exposure to the Canadian healthcare system were unable to acclimatise to Canadian cultural and practice norms [60,72].

Some IMGs resorted to suboptimal preparation strategies, such as learning checklists prepared by other IMGs who had taken the examination in the past, and memorising rote phrases to try to mitigate for the unfamiliar simulated patient encounters [62].

Seven sources reported that IMGs felt the examinations to be unrealistic and not adequately reflective of the required competencies [3,53,66,78,82,83,87]. IMGs perceived that clinical assessments were fake and described how they changed their consultation style in exams by *"performing" "acting"* and *"playing the game",* which they believed would improve their chances of examination success: [64,78,79].

In contrast to that, IMGs who took a simulated surgery OSCE type examination (general practice) believed that the case mix was realistic, but this examination has now been discontinued in favour of workplace-based assessment [87]. IMGs reported that workplace-based assessment was more realistic than OSCEs [61,88] and was more acceptable because it assessed the types of activities routinely performed in practice [75].

**Bureaucracy**

*"Where [can I] get my birth certificate? My city is under the rule of ISIS now, how can they think that I can go there and bring this paper?"* IMG – [80].

IMGs described many bureaucratic hurdles related to examinations [3,55,57,60,67,80,81,84]. "*Constantly changing rules*" and "*unrealistic*" deadlines were cited as significant challenges in the application process [57].

Five of the sources ([49,51,57,76,80] specifically reported on the experiences of refugee doctors, who found the impact of bureaucracy was particularly burdensome.

Having left behind the problems of their homelands they did not expect to find so many difficulties in their new countries [51]. The loss of their prior status compounded their difficulties [49]. as articulated by one refugee IMG:

### Feedback

*"I think the process is complicated and very difficult. The clinical examination doesn't provide proper feedback, and nobody really knows what is being assessed"* – IMG – [3].

IMGs frequently criticised the insufficient feedback received after failing examinations, which hindered their ability to prepare and improve for subsequent attempts [3,54,66,72]. In contrast, IMGs undergoing workplace-based assessment were more satisfied with the feedback they received [61,71,88,89]. Similarly, IMGs who were assessed by the Induction and Refresher Scheme simulated surgery assessment for GP returners (incorporating the International GP recruitment scheme) were satisfied with the feedback they received. In this scheme, free-text feedback from examiners and simulated patients was given to candidates as an *"educational prescription"* to allow them to improve their performance in subsequent attempts [87]. In contrast one study highlighted the fact that IMGs were not always used to receiving individualised feedback on their performance and having failed their examination they found the feedback that they received overly "critical and painful" [74].

### Perception of fairness

*"… [as] an overseas trained doctor …you are assumed incompetent, and you have to prove your competence. Whereas if you're a local graduate you're assumed competent, and then you just have to do the bare minimum"* – IMG – [81].

Several sources documented IMGs' perception of discrimination in relation to assessment [3,35,52,55,60,75,78,79,81]. IMGs felt that they were seen as *"less than"* locally trained doctors [60] and that they needed to work harder than locally trained graduates to be noticed [35,78]. IMGs believed that examiner preferences could have an unfair effect on their examination results [3,53,71,75,79,81,89]. They described feeling judged by the clothes they wore [35] and in one Canadian study an IMG had been advised not to wear to hijab to the assessment to try to mitigate against possible examiner bias [60]. IMGs also believed that having an accent that was recognisably foreign would bias the examiners against them [35,52,55,66,78].

Poor standardization among examiners was felt to be an issue [71,88]. Examiners assessing their own area of expertise and having unrealistically high expectations was mentioned as being unfair [71,88]. One IMG said that linguistic demands were too high in the examination and believed that this reflected preferences from the examiners rather than the realistic needs for the job [53].

IMGs described having to *"fight"* to get supervision and time off to attend examinations and feeling discriminated against when they were refused entry to courses that were restricted to current trainees [54,79].

There were many references to test requirements being considered unfair [3,52,53,57,60,62,64,67,72,75,80,81,83]. For example, an intensive care specialist was unhappy that he needed to be examined in multiple unrelated specialties in the AMC Clinical exam [3]. Some IMGs in Australia felt it was unfair that other IMGs from certain countries such as the

 

UK and Ireland were exempt from licensing examinations [55,81]. In one source, IMGs believed it was discriminatory that they needed to pass the required examinations based on Canadian medical practices and standards without receiving any training or observation opportunities; they believed that it was evidence of systemic prejudice that their education, training, and experience were devalued, and that they were "demoted to the level of graduating Canadian medical students" [60].

Three sources which related to workplace-based assessment found that the fact that the assessments were being carried out by their supervisors, who knew them already, could have a big impact [75,88,89]. In some cases, this was seen as positive and was cited as reducing anxiety levels [89]. In other cases, there was a worry that if the trainer and trainee did not have a good relationship, their performance would not be assessed fairly [75]. Other forms of assessment such as the Annual Review Competency Progression [79], the Induction and Refresher Scheme-Simulated Surgery [87] and an internal Medicine OSCE delivered to both final year medical students and IMGs in Chile [86] were also considered by IMGs to be fair.

The role of simulated patients in clinical assessments was problematic for some IMGs, who had not experienced this before (61). One study reported that having an assessment with real patients (in workplace-based assessment) was preferable to assessment with simulated patients [61].

The categories and subcategories found in this scoping review are documented in more detail in S4 Appendix.

### What are the gaps in the literature?

The secondary research question aimed to identify the gaps in the literature relating to our knowledge and understanding of international medical graduates' experiences of clinical postgraduate and licensing medical examinations.

Knowledge gaps identified by the authors of the sources included in the scoping review were: the impact of age, country of primary medical qualification and years since last in practice [87]; the impact of lower levels of language proficiency on IMG assessment [72]; and the impact of gender [49,52,57,80] on IMGs' experiences of assessment. One study suggested documenting the longitudinal experiences of IMGs from when they first applied for licensure to the end of the process – whether that was successfully becoming licensed to practice or the IMG withdrawing from the process [52].

Terry et al [89] suggested that there was a need for a comparison between OSCE based assessment versus WBA for evaluation of IMGs. Siriwardena et al [75] suggested that there should be an evaluation of the reliability of WBA as a means of assessing IMG readiness for registration. An evaluation of the impact of bridging programmes and structured observership on IMGs' experiences of clinical assessment was proposed [72] as was linguistic and content analysis of test items where there is differential attainment to reduce possible test bias [53]. Finally, while most sources looked at the issues that cause IMGs to fail assessments, one suggested that more research was needed that focused on predictors of success rather than predictors of failure [73].

## Discussion

This scoping review is the first to map the experiences of international medical graduates in relation to licensing and postgraduate examinations. Forty-four sources were included in the final review, drawing on the experiences of IMGs from 55 named countries spanning all 6 WHO world regions. The findings reveal multiple disparate issues that IMGs have experienced in the context of their clinical assessments. Each IMG's journey is unique to their own individual circumstances and is shaped by a complex interplay of internal personal barriers, external social issues, and systemic barriers embedded within the assessment processes themselves.

One of the most significant findings from this review is that IMGs encounter profound communication difficulties that extend far beyond basic language proficiency and involve nuanced cultural and professional communication styles. These difficulties mirror those reported by other international healthcare professionals, including a study from Canada of internationally educated nurses undertaking a bridging programme which found that the nurses struggled with language and communication skills even when they believed that they were already proficient in English [90]. While IMG trainees have

been shown to benefit from specific communications skills and linguistic interventions their use is often limited to small scale pilot studies [91–93]. Therefore, without addressing these communication barriers, clinical assessments risk unfairly penalising IMGs on cultural differences rather than a lack of clinical competence.

The findings highlight the emotional toll that high-stakes clinical assessment exacts on IMGs, who report anxiety, loss of confidence, frustration, anger and even depression, frequently compounded by financial and family responsibilities. This distress stems from multiple sources identified in the literature, including unfamiliarity with the types of assessments used and difficulty finding information about assessments. A recently published systematic review and meta-ethnography [94] highlighted that, while IMGs are known to experience significant stress due to issues like loss, adjustments, disorientation and bias, the mental health of IMGs is under-researched. It is important to note the complexity surrounding mental health: while this review found IMGs reporting high anxiety and distress, a few studies have conversely shown IMGs tend to experience less burnout than locally trained graduates [95–97]. West et al found that greater educational debt was associated with physician burnout [95]. In West's study, IMGs were less likely than locally trained graduates to have significant debt, and they postulated that this might be a reason for less burnout in IMGs [95]. Another possible interpretation of this could be that IMGs feel vulnerable in their positions and chose not to disclose burnout. Recognising and addressing this hidden psychological burden of assessments is crucial, as it impacts not only the fairness of the assessments but also the long-term wellbeing and retention of the IMG physician workforce.

This review found that supportive relationships with trainers, colleagues, fellow IMGs and locally trained graduates were consistently identified as vital for IMG success in clinical assessment. This reinforces the findings of a report commissioned by the GMC in 2019, which emphasised the importance of having inspirational senior colleagues, supportive trainers and access to a network of peers in relation to success in training [98]. Kehoe et al [99] make the point that IMGs' training and supervision needs vary depending on their prior training. They recommend addressing training needs by developing a continuing support system including a designated trainer and access to a peer "Buddy", with inbuilt training for both the trainers and "Buddies". This need for proactive integration is echoed by Haddad et al [100] who found that social connections, especially with host country natives, are pivotal to helping IMGs develop intercultural competence and that educational supervisors, organisations and locally trained doctors should help by "opening the door". A recent scoping review suggested using tailored interventions for IMGs, including better induction programmes, and continued social supports to help them transition to working in their host countries [101]. This points to the value of tailored interventions, including better induction programmes and continued social supports, to help IMGs transition successfully [98].

A critical concept emerging from this review was the perception among IMGs that clinical assessments were not only unfair but, in some cases, discriminatory. These subjective experiences provide a crucial perspective that aligns with the well-documented, objective data on differential attainment in medical assessments [20,79]. The finding that IMGs perceived discrimination within assessment processes is concerning and aligns with existing evidence, including reports of discrimination related to accent, race, language mastery and gender [102,103]. A recent Australian study found that 59% of IMGs had experienced discrimination in the previous 5 years [102] while a UK study reported discriminatory experiences related to accent, race, mastery of the English language, culture, religion, country of birth, name, and gender (being female) [103]. To move from identifying these perceptions to actively addressing potential inequities, objective analytical methods are essential. Tools developed by researchers like Tavakol et al are useful to identify and address potential fairness issues among different student demographics [104]. Where differential attainment exists, licensing and postgraduate assessment bodies must rigorously investigate its underlying causes and address any identified bias or discrimination. Failure to do so may expose these bodies to legal challenges as demonstrated by the case involving the Royal College of General Practitioners Clinical Skills Assessment in the UK [105].

Beyond the gaps explicitly identified by the included study authors, our scoping review revealed several critical evidence gaps. While a large body of research exists on IMG relocation and their quantitative pass/fail rates, our review reveals a scarcity of sources dedicated to the subjective experience of the assessment process itself. Furthermore only 18 of the 44

(41%) included sources centred on IMG assessment experience [35,47,50,53,54,58,62,70,71,73–75,77,78,86–89]. The other sources had a broader scope such as IMG perceptions about residency training experience [59] and IMG integration [49,76]. Thus, more research, which focuses in more depth on how IMGs experience assessment and how the outcomes of the assessment affect them emotionally and personally, is required.

General practice was the most commonly studied specialty with very little research available from many other specialties suggesting that other postgraduate training colleges should consider replicating the work that has been done on the RCGP CSA, to learn IMGs' perspectives on examinations across the field of medical specialties. From a licensing perspective, the only large-scale reports commissioned by medical licensing authorities came from the UK General Medical Council and the Australian Medical Association, which points to the need for other medical licensing authorities to commission similar large-scale research.

This review also reveals a significant geographical bias in the existing literature. Most of the available research came from a small number of countries – specifically the UK, Australia and Canada with very little from the many other OECD countries which rely on the work of IMGs. This geographic concentration limits the generalisability of the findings. Expanding the international research landscape is therefore essential for developing a truly global, in-depth understanding of medical migration and IMGs' experiences of assessment, which is crucial for developing targeted support interventions, designing fairer assessments, improving IMG well-being overall.

## Strengths and limitations

To our knowledge, this is the first scoping review to comprehensively map the experiences of IMGs with clinical competency assessments. Key strengths include methodological rigor [42,43,106] following established frameworks, a comprehensive search strategy encompassing grey literature and a study team with experience as IMGs (both current and former) and medical assessors, which provided a nuanced perspective that enhanced the interpretation of the findings. The date restrictions of 2009–2025 were used to ensure that the sources were relevant for the methods of assessment in current use.

The primary limitation is a potential linguistic bias: although no language restrictions were applied, most of the sources in review were written in English. Only one of the sources included in the final review was in a language other than English so the findings may not be generalizable to non-anglophone countries. As is standard for scoping reviews, we did not conduct a formal quality appraisal of the included sources, as the goal was to map the breadth of available evidence rather than assess its methodological quality. This means that the findings from all sources are presented with equal weight, regardless of study design or methodological rigour.

## Conclusions

This review synthesises the evidence regarding IMGs' experiences of clinical competency assessments, revealing a complex mixture of personal, social, and systemic challenges. Our findings present a critical dilemma for regulatory and postgraduate training bodies who bear a dual responsibility in ensuring that the doctors they license and certify are safe practitioners whilst ensuring that the examinations they oversee are fair to all doctors undertaking them. Given the increasing reliance of many counties on the work of IMGs, this issue needs to be fully investigated, including documenting the experiences and perspectives of IMGs themselves.

The insights from this review can directly inform the development of fairer assessment methodologies and more effective, targeted support interventions for IMGs. This review not only consolidates the current state of knowledge but also highlights key gaps in the evidence providing important directions for future research.

Based on the findings of this scoping review, we suggest that licensing organisations and postgraduate training bodies should introduce individualised supports for IMGs including communication and cultural adaptation training, opportunities for supported observership, support for integration with locally trained graduates, training for trainers, and better access to

information regarding regulations and assessments. Most critically, if we want to fully understand the reasons for differential attainment, we must listen to the voices of those affected – in this case the IMGs themselves – and respond their concerns. By creating fairer and more supportive assessment pathways, healthcare systems can harness the full potential of their IMGs, ensuring their valuable skills and diverse experiences are retained for the benefit of all patients.

## Supporting information

**S1 Appendix. Preferred Reporting Items for Systematic reviews and Meta-Analyses extension for Scoping Reviews (PRISMA-ScR) Checklist.**
(DOCX)

**S2 Appendix. Search strategy for PubMed (National Library of Medicine) and sample grey search strategy.**
(DOCX)

**S3 Appendix. Data Extraction Tool.**
(DOCX)

**S4 Appendix. Description of the Sources.**
(DOCX)

**S5 Appendix. Findings: Categories and sub-categories.**
(DOCX)

## Acknowledgments

We would like to acknowledge the contribution of Virginia Conrick, UCC Librarian, who assisted with devising the search strategy.

## Author contributions

**Conceptualization:** Helen Hynes, Nora McCarthy, Anél Wiese, Tony Foley, Deirdre Bennett.

**Data curation:** Helen Hynes.

**Formal analysis:** Helen Hynes.

**Investigation:** Helen Hynes, Nora McCarthy, Anél Wiese.

**Methodology:** Helen Hynes, Anél Wiese, Catherine Sweeney, Tony Foley, Deirdre Bennett.

**Project administration:** Helen Hynes.

**Supervision:** Nitin Gambhir, Tony Foley, Deirdre Bennett.

**Validation:** Nitin Gambhir, Tony Foley, Deirdre Bennett.

**Writing – original draft:** Helen Hynes.

**Writing – review & editing:** Helen Hynes, Nora McCarthy, Anél Wiese, Nitin Gambhir, Tony Foley, Deirdre Bennett.

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
