## [Decision Letter · Decision Letter 0]

5 Apr 2026

PONE-D-25-62031International medical graduates’ experiences of clinical competency assessment in postgraduate and licensing examinations: A scoping reviewPLOS One

Dear Dr. Hynes,

Thank you for submitting your manuscript to PLOS ONE. After careful consideration, we feel that it has merit but does not fully meet PLOS ONE’s publication criteria as it currently stands. Therefore, we invite you to submit a revised version of the manuscript that addresses the points raised during the review process.

Please make sure to carefully review all of reviewers' comments and then respond to each comment.  ==============================

If applicable, we recommend that you deposit your laboratory protocols in protocols.io to enhance the reproducibility of your results. Protocols.io assigns your protocol its own identifier (DOI) so that it can be cited independently in the future. For instructions see: https://journals.plos.org/plosone/s/submission-guidelines#loc-laboratory-protocols. Additionally, PLOS ONE offers an option for publishing peer-reviewed Lab Protocol articles, which describe protocols hosted on protocols.io. Read more information on sharing protocols at . Additionally, PLOS ONE offers an option for publishing peer-reviewed Lab Protocol articles, which describe protocols hosted on protocols.io. Read more information on sharing protocols at https://plos.org/protocols?utm_medium=editorial-email&utm_source=authorletters&utm_campaign=protocols..

As the corresponding author, your ORCID iD is verified in the submission system and will appear in the published article. PLOS supports the use of ORCID, and we encourage all coauthors to register for an ORCID iD and use it as well. Please encourage your coauthors to verify their ORCID iD within the submission system before final acceptance, as unverified ORCID iDs will not appear in the published article. *Only* the individual author can complete the verification step; PLOS staff the individual author can complete the verification step; PLOS staff *cannot* verify ORCID iDs on behalf of authors.verify ORCID iDs on behalf of authors.

We look forward to receiving your revised manuscript.

Kind regards,

Kento Sonoda, MD

Academic Editor

PLOS One

Journal Requirements:

2. Please upload a new copy of Figure 1 as the detail is not clear. Please follow the link for more information:  https://journals.plos.org/plosone/s/figures..

Additional Editor Comments:

Thank you for submitting your article to PLOS One. Overall, your paper is well-written and organized. Please carefully review comments from two reviewers. As reviewer 1 noted, please clarify your study selection process (inclusion and exclusion criteria) more. Consider creating a figure to illustrate your selection process with the relevant number of studies each phase.

Reviewer's Responses to Questions

**Comments to the Author**

1. Is the manuscript technically sound, and do the data support the conclusions?

Reviewer #1: Yes

Reviewer #2: Yes

2. Has the statistical analysis been performed appropriately and rigorously? 

Reviewer #1: No

Reviewer #2: N/A

3. Have the authors made all data underlying the findings in their manuscript fully available?

Reviewer #1: No

Reviewer #2: Yes

4. Is the manuscript presented in an intelligible fashion and written in standard English?

Reviewer #1: Yes

Reviewer #2: Yes

5. Review Comments to the Author

Reviewer #1: Review Comments to Authors

The authors have done well by conducting a scoping review on such an important topic. However, some issues have been identified during the review process that require the authors to address.

Abstract

Line 52: Revise ‘conclusions’ to ‘conclusion’

Methodology

Line 135: The process followed Arksey & Omalley’s Framework. However, line 32 under the abstract indicated that the Joana Briggs Institute was employed in this scoping review. Please revise this anomaly in the abstract and under the methodology section. I suggest the authors state this clearly, as it forms the heart of this scoping review.

Line 166: Study selection.

• Please, the content under this section does not indicate the number of publications identified from the initial search. After deduplication (how many duplicates were removed), how many were removed after abstracts and titles were screened? What were the reasons for their removal? These review comments are relevant mainly because the PRISMA Flow Diagram was illegible due to poor image quality. I suggest that the authors revise this section, as it shows the validity and reliability of the scoping review.

• I suggest that the authors provide a table showing the published materials considered after removing duplicates, clearly indicating those included and excluded, as well as the reasons for exclusion.

Line 183: A pilot test was conducted to trial the inclusion criteria.

Line 150: Identifying relevant sources for inclusion: In the abstract, the authors indicated that there was no language limitation. However, the methodology section did not mention this criterion. I suggest that the authors revise this section to indicate whether some peer-reviewed articles (number specified) were identified in other languages (language specified) and the decision made regarding such articles. This information needs to be included in the ‘description of the sources.’

Line 432-425: Please, revise for clarity.

Line 471: What does ‘this’ mean in that quote/extract?

Line 510: Revise the quote for clarity. Consider ‘Where can I get my birth certificate?’

Line 591: Revise (Tipton, 2011) to be consistent with the preferred referencing style of the journal.

Line 622: Please, do not start a sentence with a number (44), revise to Forty-four (44)

Line 885: I suggest the authors check this reference to confirm the presence of the dot between physician and Le (Physician . Le)

Supporting Information

S1: Figure 1, Scoping Review PRISMA Flow Diagram is blurred. Hence, difficult to appreciate. I suggest that the authors upload a clearer figure.

S4: Description of Sources

I suggest the authors used descriptive statistics to describe the sources under the following headings:

• Year

• Country of publication

• Type of study

• Examination

• Specialty

Reviewer #2: This is a well-conducted scoping review that makes a valuable contribution to the literature on international medical graduates' (IMGs) experiences of clinical competency assessment. The topic is important, the methodology is clearly described, and the findings are presented in a structured and accessible manner. I recommend acceptance subject to the following minor revisions.

Data extraction process

The authors note that while the data extraction tool was piloted by three reviewers working in pairs, the remaining extraction was carried out solely by the principal investigator (HH). For a scoping review aiming for methodological rigour, it would strengthen the paper if the authors could clarify what steps were taken to verify the accuracy and consistency of the single-reviewer extraction, or acknowledge this more explicitly as a limitation.

6. PLOS authors have the option to publish the peer review history of their article (what does this mean?). If published, this will include your full peer review and any attached files.). If published, this will include your full peer review and any attached files.

.

Reviewer #1: No

Reviewer #2: No

---

## [Author Response · Author response to Decision Letter 1]

6 Apr 2026

Dear Dr Sonoda and Reviewers,

Many thanks for reviewing our article – “International medical graduates’ experiences of clinical competency assessment in postgraduate and licensing examinations: A scoping review” [PONE-D-25-62031] and for the opportunity to resubmit with revisions.

We have considered each of the points made by the reviewers and have made the following amendments in response to same.

Reviewer #1

Abstract Line 52: Revise ‘conclusions’ to ‘conclusion’

Response: This has been revised as suggested.

Methodology Line 135: The process followed Arksey & Omalley’s Framework. However, line 32 under the abstract indicated that the Joana Briggs Institute was employed in this scoping review. Please revise this anomaly in the abstract and under the methodology section. I suggest the authors state this clearly, as it forms the heart of this scoping review -

Response: Thank you. This has been clarified by removing the reference to using Joanna Briggs methodology - see line 32 of the abstract.

Line 166: Study selection. • Please, the content under this section does not indicate the number of publications identified from the initial search. After deduplication (how many duplicates were removed), how many were removed after abstracts and titles were screened? What were the reasons for their removal? These review comments are relevant mainly because the PRISMA Flow Diagram was illegible due to poor image quality. I suggest that the authors revise this section, as it shows the validity and reliability of the scoping review.

• I suggest that the authors provide a table showing the published materials considered after removing duplicates, clearly indicating those included and excluded, as well as the reasons for exclusion.

Response: The Prisma flow diagram has been re-saved. It is now legible and shows the details required.

Line 183: A pilot test was conducted to trial the inclusion criteria.

Response: A description has been added of how this took place.

Line 150: Identifying relevant sources for inclusion: In the abstract, the authors indicated that there was no language limitation. However, the methodology section did not mention this criterion. I suggest that the authors revise this section to indicate whether some peer-reviewed articles (number specified) were identified in other languages (language specified) and the decision made regarding such articles. This information needs to be included in the ‘description of the sources.’

Response: This information has now been added to the section on "description of the sources".

Line 432-425: Please, revise for clarity.

Response: Thank you. This has been revised for clarity

Line 471: What does ‘this’ mean in that quote/extract?

Response: "This” refers to OSCE type assessment - this has now been clarified in the text.

Line 510: Revise the quote for clarity. Consider ‘Where can I get my birth certificate?’

Response: Thank you. This has been edited as advised.

Line 591: Revise (Tipton, 2011) to be consistent with the preferred referencing style of the journal.

Response: Edited as suggested. Thank you for spotting this.

Line 622: Please, do not start a sentence with a number (44), revise to Forty-four (44)

Response: Edited. Thank you.

Line 885: I suggest the authors check this reference to confirm the presence of the dot between physician and Le (Physician . Le)

Response: Amended. Thank you.

Supporting Information S1: Figure 1, Scoping Review PRISMA Flow Diagram is blurred. Hence, difficult to appreciate. I suggest that the authors upload a clearer figure. The Prisma flow diagram has been re-saved. It is now legible and shows the details required.

S4: Description of Sources I suggest the authors used descriptive statistics to describe the sources under the following headings:

• Year

• Country of publication

• Type of study

• Examination

• Specialty

Response: The section on description of the sources has been amended to include descriptive statistics as suggested.

Reviewer #2

Data extraction process The authors note that while the data extraction tool was piloted by three reviewers working in pairs, the remaining extraction was carried out solely by the principal investigator (HH). For a scoping review aiming for methodological rigour, it would strengthen the paper if the authors could clarify what steps were taken to verify the accuracy and consistency of the single-reviewer extraction, or acknowledge this more explicitly as a limitation.

Response: Thank you. The remaining data extraction was carried out by the principal investigator (HH) and was reviewed for accuracy and consistency by the other members of the research team. This has now been clarified in the text. Please see lines 208 -209 in the section on Data Extraction in the Revised Manuscript with tracked changes

Thank you sincerely for your consideration of this manuscript.

Warm Regards,

Helen Hynes

---

## [Decision Letter · Decision Letter 1]

14 Apr 2026

International medical graduates’ experiences of clinical competency assessment in postgraduate and licensing examinations: A scoping review

PONE-D-25-62031R1

Dear Dr. Hynes,

We’re pleased to inform you that your manuscript has been judged scientifically suitable for publication and will be formally accepted for publication once it meets all outstanding technical requirements.

An invoice will be generated when your article is formally accepted. Please note, if your institution has a publishing partnership with PLOS and your article meets the relevant criteria, all or part of your publication costs will be covered. Please make sure your user information is up-to-date by logging into Editorial Manager at Editorial Manager® and clicking the ‘Update My Information' link at the top of the page. For questions related to billing, please contact  and clicking the ‘Update My Information' link at the top of the page. For questions related to billing, please contact billing support..

Kind regards,

Kento Sonoda

Academic Editor

PLOS One

Additional Editor Comments (optional):

Thank you for addressing the reviewers' comments. Thank you for submitting your paper to our journal.

Reviewers' comments:

Reviewer's Responses to Questions

**Comments to the Author**

1. If the authors have adequately addressed your comments raised in a previous round of review and you feel that this manuscript is now acceptable for publication, you may indicate that here to bypass the “Comments to the Author” section, enter your conflict of interest statement in the “Confidential to Editor” section, and submit your "Accept" recommendation.

Reviewer #2: All comments have been addressed

Reviewer #3: All comments have been addressed

2. Is the manuscript technically sound, and do the data support the conclusions?

Reviewer #2: Yes

Reviewer #3: Yes

3. Has the statistical analysis been performed appropriately and rigorously? 

Reviewer #2: N/A

Reviewer #3: N/A

4. Have the authors made all data underlying the findings in their manuscript fully available?

Reviewer #2: Yes

Reviewer #3: Yes

5. Is the manuscript presented in an intelligible fashion and written in standard English?

Reviewer #2: Yes

Reviewer #3: Yes

6. Review Comments to the Author

Reviewer #2: The authors well responded to the issue which I pointed out.

Reviewer #3: The authors well responded to the issues.

7. PLOS authors have the option to publish the peer review history of their article (what does this mean?). If published, this will include your full peer review and any attached files.). If published, this will include your full peer review and any attached files.

.

Reviewer #2: No

Reviewer #3: No

---

## [Editor Report · Acceptance letter]

PONE-D-25-62031R1

PLOS One

Dear Dr. Hynes,

I'm pleased to inform you that your manuscript has been deemed suitable for publication in PLOS One. Congratulations! Your manuscript is now being handed over to our production team.

Kind regards,

on behalf of

Dr. Kento Sonoda

Academic Editor

PLOS One